

# Biogeochemical versus ecological consequences of modeled ocean physics

Sophie Clayton[1, 2], Stephanie Dutkiewicz[1], Oliver Jahn[1], Christopher Hill[1], Patrick Heimbach[1, 3], and Michael J. Follows[1]

[1]Department for Earth, Atmospheric, and Planetary Sciences, Massachusetts Institute of Technology, Cambridge, MA, USA
[2]School of Oceanography, University of Washington, Seattle, WA, USA
[3]Institute for Computational Engineering and Sciences and Jackson School of Geosciences, The University of Texas at Austin, Austin, TX, USA

*Correspondence to:* Sophie Clayton (sclayton@uw.edu)

**Abstract.** Regional and idealized modeling studies have shown that increasing the physical resolution of biogeochemical models to include mesoscale and submesoscale dynamics can result in both increases and decreases in phytoplankton biomass and primary production, as well as changes in phytoplankton community structure. Here we present a systematic study of the differences generated by coupling the same ecological-biogeochemical model to a $1^o$, coarse-resolution, and $1/6^o$, eddy-permitting, global ocean circulation model. Surprisingly, we find that the modeled phytoplankton community is largely unchanged, with the same phenotypes dominating in both cases. Conversely, there are large regional variations in integrated primary production, phytoplankton and zooplankton biomass. In the subtropics, mixed layer depths are, on average, deeper in the eddy-permitting model, resulting in higher nutrient supply driving increases primary production and phytoplankton biomass. In the higher latitudes, deeper spring mixed layer depths in the eddy-permitting model result in increased light limitation during the spring bloom. Counter-intuitively, this does not drive a decrease in phytoplankton biomass, but is reflected in decreased primary production and zooplankton biomass. We explain these similarities and differences in the model using the framework of resource competition theory, and find that they are the consequence of changes in the regional and seasonal nutrient supply and light environment, mediated by differences in the modeled mixed layer depths. Although previous work has suggested that complex models may respond chaotically and unpredictably to changes in forcing, we find that our model responds in a predictable way to different ocean circulation forcing, despite its complexity.

## 1 Introduction

Ocean general circulation models have proved an invaluable tool for studying the role of phytoplankton in the global biogeochemical cycles of climatically important elements. Recent advances have resulted in ever higher resolution physical models of the ocean circulation (Menemenlis et al., 2008), and more complex ecological models incorporating larger numbers of phytoplankton functional groups, and even individual phytoplankton phenotypes (Follows and Dutkiewicz, 2011). This trend for increasing resolution and complexity is aimed at creating model systems which incorporate some of the complexity seen in reality, with the hope of better resolving biogeochemical processes.



The physical framework of an ocean model is fundamental to accurately modeling biogeochemical cycles and phytoplankton ecology (Doney, 1999; Anderson, 2005). Observations have shown that phytoplankton biomass and community structure have characteristic temporal and spatial scales corresponding most closely with the oceanic mesoscale and submesoscale (Platt, 1972; Strass, 1992; Abbott and Letelier, 1998; Doney et al., 2003; Cotti-Rausch et al., 2016). However, most global ocean

models incorporating biogeochemical and ecological processes do not resolve scales less than ~1$^o$. In the ocean, the characteristic temporal and spatial scales of biology coincide with those of mesoscale and submesoscale physical dynamics. Spall and Richards (2000) showed in a high resolution model of an unstable frontal jet that spatial heterogeneity in primary production occured on scales of a few to 10s of kms, and that primary production could increase locally by up to 100%. Coarse resolution global biogeochemical models do not resolve these dynamics. Lévy (2008) and Mahadevan (2016) review the consequences

of this for biogeochemical models. Previous studies, have found that neglecting to resolve the mesoscale could result in errors of up to 30% in the estimates of primary production (Lévy et al., 1998; Oschlies and Garçon, 1998; Mahadevan and Archer, 2000; McGillicuddy et al., 2003). Furthermore, Lévy et al. (2001) found discrepencies of up to 50% in integrated primary production comparing a coarse resolution model with one that resolved submesoscale dynamics. These studies found that, in the oligotrophic subtropical gyres, mesoscale (and submesoscale) dynamics drove an increased nutrient supply to the surface

mixed layer, which enhanced rates of primary production. In the subpolar gyres, eddies appear to have a different effect on ocean biology, McGillicuddy et al. (2003) found in a 0.1$^o$ resolution model of the North Atlantic that mesoscale processes drove a geostrophic adjustment to deep winter convection, which reduced nutrient supply. However, nutrients are less likely to be limiting than light in the subpolar gyres, so this may also have a positive effect on rates of primary production. Although these previous studies predict the role of the mesoscale in modulating biological and biogeochemical responses in different

regional settings, it is unclear what the integrated effect will be globally, and what downstream effects might result.

The above studies, although they resolved higher resolution physics, typically employed rather simple biogeochemical models incorporating only one or two phytoplankton functional types. However, marine microbial communities are known to be incredibly diverse, and this diversity plays an important role in mediating global biogeochemical cycles. Thanks to the continuing expansion of computing resources, diversity has been included in global biogeochemical models which resolve several

phytoplankton functional groups (Chai et al., 2002; Gregg et al., 2003; Quéré et al., 2005), and even several tens of phytoplankton phenotypes within mutlipe functional groups have been developed (Follows et al., 2007; Ward et al., 2012). It is unknown whether a change in physical resolution will result in any changes in the emergent community structure of one of these diverse models. Sinha et al. (2010) found that an intermediate-complexity ecosystem model which resolved 5 phytoplankton functional types, run with two different physical models at similar (coarse) resolution, could result in regional changes in the

modeled phytoplankton communities. However, it is unclear whether the emergent community resulting from a more complex ecosystem model will be more or less robust to changes in the physical forcing.

Here we present a process study that explores the effect of refining the physical resolution on a global, diverse ecosystem model which incorporates 78 distinct phytoplankton phenotypes. We explore both the effect on the bulk biogeochemical properties and the community structure of the model solutions. The objective of this study is not to assess which model performs





best with respect to reality, but to examine how changes in the resolution and parameterization of subgridscale processes of the model domain alter the emergent biogeochemical and ecological properties of this diverse ecosystem model.

## 2  Method

This study is based upon numerical simulations of global ocean circulation, biogeochemical cycles, and diverse phytoplankton populations. We have employed the Massachusetts Institute of Technology General Circulation Model (MITgcm; Marshall et al., 1997) and biogeochemical and ecological model components as detailed in Dutkiewicz et al. (2009). Below, we describe these physical circulation and biogeochemical-ecological models in more detail.

### 2.1  Physical Model Configurations

We used two physical model configurations developed by the Estimating the Circulation and Climate of the Oceans (ECCO) project. Both span the period 1992 - 1999, and were constrained to be consistent with observed altimetry and hydrography. The high resolution ECCO2 configuration, with an effective resolution of $1/6^o$, and horizontal grid dimensions of ~18 km (ECCO2; Menemenlis et al., 2008), is referred to as eddy-permitting, as it resolves large eddies at low latitudes, but remains below the first baroclinic Rossby radius at high latitudes (Chelton et al., 1998). We compare this high resolution configuration with the ECCO-Global Ocean Data Assimilation Experiment state estimate (ECCO-GODAE, also referred to as ECCO version 3 in the lineage of ECCO production solutions; Wunsch and Heimbach, 2007, 2013), which has a coarser grid resolution of $1^o$.

A core focus of this study is comparing and contrasting the modeled ecological and biogeochemical behavior in a mesoscale eddy-permitting model (ECCO2) with a non-eddying model (ECCO-GODAE). Accordingly we limit our analysis latitudinally to the region between $60^o$S and $60^o$N. Within this latitudinal band, the first baroclinic radius of deformation (Chelton et al., 1998) is larger than the ECCO2 model horizontal resolution, and so the ECCO2 model admits corresponding mesoscale eddy dynamics. North and south of this latitude line, mesoscale eddies are not well resolved in either model, and so neither model is in a so-called large-eddy regime (Smagorinsky, 1963). In the excluded high latitude regions the two physical model mixed layers differ systematically. These differences result from the absence of parameterized mesoscale eddy dynamics in the ECCO2 model (Danabasoglu et al., 1994). This contrast in mixed layer depth, rather than behaviours due to mesoscale eddies, sets the differences in biogeochemical response seen in ice-free high latitudes. For simplicity, we will refer to the ECCO-GODAE simulation as CR, and the ECCO2 simulation as HR.

### 2.2  Ecological and Biogeochemical Model

The ecological model used in this study has previously been discussed in Follows et al. (2007) and Dutkiewicz et al. (2009). Briefly, we transport inorganic and organic forms of nitrogen, phosphorous, iron and silica, and resolve 78 phytoplankton phenotypes and two simple grazers. The biogeochemical and biological tracers interact through the formation, transformation and remineralization of organic matter. Excretion and mortality transfer living organic material into sinking particulate and dissolved organic detritus which are transpired back to inorganic form. The time rate of change in the biomass of each of the



modeled phytoplankton types, $P_j$, is described in terms of a light, temperature and nutrient dependent growth, sinking, grazing, mortality and transport by the fluid flow. Many realizations of this ecological model, coupled to the ECCO-GODAE physical circulation, have been used to study a range of ecological questions, e.g. the role of top-down controls in setting patterns of phytoplankton diversity (Prowe et al., 2012; Vallina et al., 2014), the biogeography of nitrogen-fixing phytoplankton (Monteiro

et al., 2011), and role of transport in setting patterns in phytoplankton diversity (Clayton et al., 2013).

In this study, the ecological model was initialized with seventy-eight phytoplankton phenotypes with a broad range of physiological attributes. Phytoplankton were assigned to one of two broad size classes by random draw at the initialization of the model, and a set of physiological trade-offs that reflect empirical observations were imposed accordingly. We stochastically assigned plausible values for the nutrient half-saturation constants ($\kappa_N$), light and temperature sensitivities within each phy-

toplankton functional group (*Prochlorococcus*-like, picophytoplankton, diatoms and large phytoplankton). Both the HR and CR simulations were initialized with an identical set of phytoplankton phenotypes, and initial conditions for all phytoplankton phenotypes, and in both cases the ecosystem model was forced with identical initial conditions for all variables, light forcing and dust inputs. Interactions with the environment, competition with other phytoplankton, and grazing determine the composition of the phytoplankton communities that persist in the model solutions. We then compare the solutions from both model

configurations to test the sensitivity of the ecosystem to the modeled ocean physics.

## 3  Results

Differences in the physical circulation estimates of these two configurations have previously been discussed in Clayton et al. (2013). We describe differences in some of the physical properties most directly relevant to biogeochemical processes: sea surface temperature (SST) and mixed layer depth (MLD). Although the SST patterns in both models are broadly similar, we

found local differences in SST of up to $3^oC$ in some regions (Fig. 1). The HR simulation appeared to have a slight cool bias relative to the CR simulation, which may be an indicator of enhanced upwelling and/or vertical mixing in the higher resolution configuration. There were marked regional patterns in the MLD differences between model configurations. Annual average MLDs were consistently deeper in the low latitudes in the HR simulation, whereas they tended to be slightly shallower in the mid-latitudes. As mentioned above, we exclude the high latitudes ( $> 60^o$) from our analysis.

### 3.1  Primary Production and Biomass

Both model configurations result in largely similar patterns in phytoplankton biomass and primary production, with low biomass and productivity associated with the subtropical gyres, and higher biomass and productivity found in the mid-latitudes and upwelling zones. Although globally integrated primary production is very similar between both models, there are clear regional differences (Fig. 2), with higher rates of production in the low latitudes and lower rates of production in the mid latitudes,

in the HR simulation. We find a ~20% global increase in the standing stock of phytoplankton biomass in the HR simulation, mostly accounted for by higher phytoplankton biomass between $40^oS$ and $40^oN$. The largest differences in both phytoplankton biomass and primary production are associated with the western boundary currents and the equatorial upwelling zone, with





higher values in the HR simulation. We also see decreases in phytoplankton biomass and productivity in the HR simulation associated with the boundaries between different biogeographical provinces.

## 3.2 Phytoplankton Community Structure

The ecological model represents a diverse community of phytoplankton, subdivided into different sizes and functional groups, with different temperature, light and nutrient requirements. We find that the emergent phytoplankton community structure is remarkably similar between the two simulations (Fig. 3). The low latitudes are dominated by *Prochlorococcus*-like phytoplankton, whereas the mid-latitudes are dominated by diatoms and large phytoplankton (Fig. 3a and b), with very little difference between models. Similarly, the dominant phytoplankton phenotype is also largely unchanged between model simulations (Fig. 3c and d). The only region that shows a shift in the dominant functional group and phenotype between model simulations, is the Indian Ocean, which shifts from picophytoplankton dominated in the CR simulation, to *Prochlorococcus*-like phytoplankton dominated in the HR simulation, with a corresponding shift in the dominant phenotype. Other regions where the models do not agree occur mainly at the borders between different biogeochemical provinces.

Differences in the patterns of phytoplankton biodiversity between the model simulations have previously been described in Clayton et al. (2013). Although we find an overall increase in phytoplankton biodiversity in the HR simulation, those differences are mainly driven by the persistence of more rare species in the HR simulation with respect to CR, and are tus unlikely to have a significant effect on integrated bulk biogeochemical processes.

## 3.3 Nutrients and Nutrient-Limitation

Differences in model resolution are expected to drive changes in the supply of nutrients to the surface photic zone. We do find differences in the concentration of surface macro- and micro-nutrients (Fig. 4). Nitrate and dissolved iron both show marked patterns in the difference of their surface distributions between simulations. The concentration of nitrate remains unchanged in the Atlantic, the Indian Ocean, and in some parts of the North and South Pacific (Fig. 4a). However, there is a decrease in surface nitrate concentrations in the HR simulation in the mid-latitudes of the North Pacific and the Southern Ocean, and in the subtropical and equatorial Pacific. Conversely, we see an increase in surface nitrate in the HR simulation in the Brazil-Malvinas Confluence Zone, and the region of the Subantarctic Front in the Southern Ocean. Differences in surface dissolved iron concentrations exhibit markedly different patterns (Fig. 4b). The surface concentration of dissolved iron remains unchanged between model simulations in the the subtropical and equatorial Pacific, and parts of the subtropical North Atlantic. We see a general decrease in surface iron in the HR simulation in the Subantarctic Front in the Southern Ocean and the southern part of the Indian Ocean, and an increase in dissolved iron in the mid-latitudes of the North Pacific, the equatorial Atlantic, and the subpolar North Atlantic.

In addition to assessing the differences in surface nutrient concentrations, we determined the locally limiting nutrient for both models (Fig. 5). Nitrate and dissolved iron are the spatially dominant limiting nutrients in both model simulations, with almost identical regional patterns of limitation. However, we do find small but significant regions where the dominant limiting nutrients in both simulations do not correspond (Fig. 5b), primarily located at the boundaries between biogeographical provinces.



## 4 Discussion

In this study, we have assessed the effect of explicitly representing mesoscale dynamics in a global ocean ecological-biogeochemical model. Strikingly, we find that the realized phytoplankton communities in both simulations are remarkably similar, but that there are marked regional variations in bulk ecosystem properties such as primary production and phytoplankton biomass. We also find that although the general regional patterns of nutrient limitation remain unchanged, surface concentrations of nitrate and dissolved iron can vary markedly between simulations in some regions, but are almost completely unchanged in others. Why do we see such marked changes in the distribution of biogeochemical properties of the models, but not in the modeled phytoplankton community?

Ultimately, any differences in the biogeochemical and ecological properties between the two model solutions occur either because there are differences in the local physical fields (e.g. MLDs and transport), or because physical features of the ocean circulation such as the western boundary currents and gyre boundaries are realized in different locations. We identify three main regions that are affected in different ways by the addition of mesoscale dynamics: the tropical and subtropical regions, the northern hemisphere subpolar gyres, and the boundaries between biogeographical provinces. We explore the linkages between the physical and biological components of the system to put these differences into context.

### 4.1 Tropical and subtropical regions

In the low latitudes, we found that the addition of mesoscale dynamics resulted in an overall deepening of the annual mean mixed layer depth (Fig. 1). We found an increase in phytoplankton biomass and primary production, but the dominant phytoplankton functional group and phenotypes remained unchanged. The increase in primary production and phytoplankton biomass in the HR simulation was not entirely surprising, as it has been predicted by previous regional and idealized studies on the effect of increasing model resolution on primary production (Lévy et al., 2001; Oschlies and Garçon, 1998; Mahadevan and Archer, 2000; Spall and Richards, 2000; McGillicuddy et al., 2003). However, we may have expected a large increase in the biomass of large phytoplankton thanks to more episodic eddy-driven nutrient injections to the surface mixed layer, as observed by Benitez-Nelson et al. (2007); Brown et al. (2008). In fact, the increase in phytoplankton biomass was driven by an increase in the dominant phytoplankton functional groups, *Prochlorococcus*-like and picophytoplankton, which are both small, gleaner types. We also found that where a particular nutrient was limiting in both model simulations, its surface concentration remained unchanged.

The subtropical gyres are steady, stable regions where seasonality is low, and phytoplankton growth is nutrient-limited. In this context, we can apply the simple resource competition framework introduced by (Tilman et al., 1982) and applied to this system by (Dutkiewicz et al., 2009) to understand the simultaneous differences and similarities in the model results. We set up a simple model system, where phytoplankton growth is nutrient-limited, and balanced by a simple linear mortality term:

$$\frac{dR}{dt} = -\frac{\mu RP}{R+k} + S_N \tag{1}$$

$$\frac{dP}{dt} = \frac{\mu RP}{R+k} - mP \tag{2}$$





where $R$ is the limiting resource, $P$ is the phytoplankton biomass, $\mu$ is the maximum phytoplankton growth rate, $k$ is the resource half saturation constant, $m$ is the phytoplankton mortality rate, and $S_N$ is the rate of resource supply into the system. The steady state solution for this system is:

$$P^* = \frac{S_N}{m} \tag{3}$$

$$R^* = \frac{mk}{\mu - m} \tag{4}$$

In this framework, the steady state concentration of the limiting resource, $R^*$, is controlled not by the magnitude of the nutrient supply, but by the physiological attributes of the dominant phytoplankton phenotype. In this type of stable system, the emergent dominant phytoplankton phenotype will be the one that can draw the resource down to the lowest $R^*$ value. As we have set the phytoplankton communities to be composed of the same phenotypes in both simulations, and $R^*$ is set by the physiological traits of the phytoplankton (Dutkiewicz et al., 2009), we select for the same dominant phenotype in both simulations. Conversely, the steady state concentration of the phytoplankton, $P^*$, is a function of the resource supply, $S_N$, as well as the mortality rate. Thus, an increase in $S_N$ results in increased phytoplankton biomass, but not in a shift in the dominant phytoplankton type, as $R^*$ is only a function of the physiology of the fittest phytoplankton type. Enhanced mixing in the HR simulation drives an increase in $S_N$, but does not change the dominant phytoplankton phenotype. Rather, it results in an increase in the biomass of the dominant type, and an increase in productivity ($\mu P^*$). At the same time, the concentration of the limiting nutrient, $R^*$, remains unchanged (as seen in nitrate and dissolved iron, figure 4), but the non-limiting nutrients decrease due to increased removal by the increased productivity (Fig. 4). This may also result in a decrease in the supply of resources downstream from the subtropical gyres.

### 4.2 Northern hemisphere subpolar regions

We see a somewhat different picture in the mid latitudes, where primary production is decreased by the addition of mesoscale dynamics, but still phytoplankton biomass remains very similar in both simulations. In contrast to the subtropical gyres, these are nutrient replete regions where patterns in phytoplankton productivity and biomass are seasonally controlled, and primarily limited by light. In order to explain the differences in the annual mean distributions of phytoplankton productivity and biomass, we must examine the seasonal evolution of the system. In Fig. 7, we show the total monthly integrated primary production, phytoplankton and zooplankton biomass for the northern hemisphere mid-latitudes for both simulations. Although in both simulations the timing and magnitude in the phytoplankton biomass at the height of the spring bloom is almost the same, we see marked differences in the magnitude of primary production, and in the seasonal abundance of zooplankton. Primary production from March to August, and zooplankton abundance over most of the year are both much lower in the HR simulation. Deeper spring MLDs in the HR simulation (Fig. 6) result in increased light limitation during the onset of the spring bloom, reducing primary production in the HR simulation (Sverdrup, 1953). But then why do we not see a corresponding decrease in phytoplankton biomass in the HR simulation, and why is zooplankton biomass so much higher in the CR simulation? We hypothesize that this is, at least in part, a consequence of the "Dilution-Recoupling Hypothesis" (Behrenfeld, 2010), where top-down control of phytoplankton biomass by grazers resumes in spring with the shallowing of MLDs. We can explain this in





the context of an idealized model system, where phytoplankton growth is a function of light, and the phytoplankton are grazed on by zooplankton:

$$\frac{dR}{dt} = -\mu(I)RP + S_N + \gamma m_z Z \tag{5}$$

$$\frac{dP}{dt} = \mu(I)RP - gPZ \tag{6}$$

$$\frac{dZ}{dt} = gPZ - m_z Z \tag{7}$$

where $R$ is the resource, $P$ is the phytoplankton biomass, $\mu(I)$ is the phytoplankton growth rate which is controlled by light, $g$ is the grazing rate of phytoplankton by zooplankton, $m_z$ is the zooplankton mortality rate, $\gamma$ is a remineralization coefficient, and $S_N$ is the rate of resource supply into the system. The steady state solution for this system is:

$$R^* = \frac{S_N g}{\mu(I)m_z}(1-\gamma)^{-1} \tag{8}$$

$$P^* = \frac{m_z}{g} \tag{9}$$

$$Z^* = \frac{S_N}{m_z}(1-\gamma)^{-1} \tag{10}$$

For this system, $P^*$, the phytoplankton standing stock is controlled by the physiological traits of the zooplankton which are constant, so regardless of changes in $\mu(I)$ or $S_N$, phytoplankton biomass will remain the same. However, $Z^*$, the zooplankton standing stock is directly proportional to $S_N$, so increased nutrient supply will translate to higher $Z^*$. Applying this framework to our model results explains why the higher primary production in the CR simulation results in higher zooplankton biomass, but unchanged phytoplankton biomass. Zhang et al. (2013) find a similar response to nutrient injections at a shelfbreak front, where increases in primary production are funneled up the trophic levels and are seen in increased zooplankton biomass, but are not reflected in increased phytoplankton biomass. On a larger scale, Ward et al. (2012) found in a global, size-structured ecosystem model, that in regions of higher nutrient supply, top-down control drives in an increase in the biomass of larger plankton size classes relative to oligotrophic regions, rather than an unchecked increase in smaller size classes.

Although we can explain the decrease in primary production as a consequence of light limitation caused by deeper spring MLDs in the HR simulation, and the unchanged phytoplankton biomass as a result of top-down control, this does not account for the lower surface nutrient concentrations seen over large regions in the HR simulation (Fig. 4). We hypothesize that this could be due to the lower zooplankton biomass resulting in decreased nutrient remineralization in the surface mixed layer, but can not show this conclusively in this study. It is also possible that despite the deeper average MLDs, eddies are episodically acting to remove inorganic nutrient from the surface, as seen in McGillicuddy et al. (2003) 0.1 $^o$ model of the North Atlantic. They found that eddies constituted a net sink of nutrients from the surface in the subpolar gyre, counteracting the wind-driven upwelling of nutrients. Conversely, Mahadevan et al. (2012) show that eddies can help to initiate the spring bloom in the North Atlantic by stratifying the water column, and thus reducing light limitation of the phytoplankton. Neither of these mechanisms square entirely with our results, showing an increased MLD and decreased primary production in the eddy-permitting HR simulation. It is possible that at 1/6$^o$ resolution, we still do not resolve mesoscale eddies well enough to represent either of these competing processes.





### 4.3 Boundaries between biogeographical provinces

We found marked differences in biological fields which appear to be due to differences in the geographical extent of bio-geographical provinces between simulations. Differences in modeled phytoplankton biomass and primary production in the Northern and Southern Pacific subtropical gyres coincide exactly with geographical differences in regions defined by differ-

ent limiting nutrients in the simulations (Fig. 2). We attribute this decrease in production and biomass to a decrease in the lateral supply of nutrients from the equatorial upwelling region. This decreased lateral supply out of the equatorial region in the HR simulation is due to more local primary production in the equatorial region (Fig. 2). Although dissolved iron is drawn down to the same concentration in both simulations, primary production and phytoplankton biomass are higher, and nitrate is much lower in the HR simulation. As discussed above, resource competition theory predicts that the concentration of the

limiting nutrient, R* (in this case dissolved iron; Fig. 5) is unchanged when the phytoplankton community remains unchanged. Increased dissolved iron supply, associated with deeper MLDs would drive the increase in primary production and phytoplankton biomass observed in the in the HR simulation. However, this increase in local primary production, consuming higher levels of non-limiting macronutrients locally, reduces the Ekman transfer of non-limiting nutrients to the neighboring subtropical gyres (Dutkiewicz et al., 2005). A similar shift in biological transition zones, associated with model resolution, was found in a

regional study of the California Current System. Fiechter et al. (2014) found that increasing the resolution of their model from $1/3^o$ to $1/30^o$ resulted in a marked shift in the location of the transition between near-shore outgassing and offshore absorption of $CO_2$. These differences in gas fluxes were driven, in their study, by different patterns in nutrient upwelling and transport offshore.

### 4.4 Stability of the phytoplankton community structure

As we have discussed in the previous sections, one of the most striking results of our model comparison is that despite the difference in the modeled physics, the emergent phytoplankton communities in both simulations are almost identical. We only know of one comparable study where the response of a complex ecosystem model to different model physics has been evaluated (Sinha et al., 2010). In that study, although there was a difference in resolution between the two physical models used to force the ecosystem model ($1^o$ vs $2^o$), unlike our study, both of these physical models were too coarse to resolve

eddies. Although the physical models in the (Sinha et al., 2010) study were initialized and forced in essentially the same way, they were run out without being constrained to observations. This resulted in large differences in their model solutions for physical fields (e.g. seasonal SST, see their Fig. 3), which were used to force the ecosystem model. One very key difference in our study with respect to Sinha et al. (2010)'s earlier work, is that both of the physical models used to force our ecosystem model, ECCO-GODAE and ECCO2, were state estimates constrained to converge as closely as possible to observational data.

Although the state estimates were constrained in different ways (Green's functions vs. adjoint), and differences between them remain, compared to free-running models without any assimilation these differences are comparatively small. In the end, the differences in the ECCO state estimates used in this study should be largely a consequence of whether or not they represent mesoscale dynamics.





## 5   Conclusions

In this study we have compared and contrasted the behavior of a complex ecological-biogeochemical model when coupled to either a mesoscale eddy-permitting high resolution physical model (HR, ECCO2), or a non-eddying coarse resolution physical model (CR, ECCO-GODAE). We found that increasing the model resolution to include mesoscale dynamics does not

greatly affect the structure of our modeled phytoplankton ecosystem. However it does have a significant effect on the regional distribution of the bulk properties of the ecosystem: primary production, and phytoplankton and zooplankton biomass.

One of the most striking results of this study is the robustness of the emergent modeled phytoplankton community. We found that the dominant phytoplankton functional groups and phenotypes remained unchanged between simulations, despite differences in SST and MLDs. In contrast, we found marked regional differences in phytoplankton and zooplankton biomass,

and primary production. By applying concepts from resource competition theory (Tilman et al., 1982; Dutkiewicz et al., 2009), we can explain why in the subtropical gyres, despite increased primary production, the dominant phytoplankton phenotype and the surface concentration of the limiting nutrient remain the same in both simulations. The combination of low seasonality and low grazing pressure means that despite an increased nutrient supply, phytoplankton with the lowest $R^*$ will always be selected for in this region.

Given the complexity of our ecosystem model, which incorporates 78 individual phytoplankton types, it may seem surprising that our modeled phytoplankton community structure is so similar in both cases. This presents interesting implications for marine biogeochemical and ecological modeling. It is clear that accuracy in the representation of the physical dynamics of the environment is necessary for effectively modeling ocean biogeochemistry. The higher taxonomic resolution of our ecological model may in fact allow for subtler gradations of change in the phytoplankton community when the environment is changed,

whereas in coarser ecological models, regime shifts could easily result due to larger differences between modeled phenotypes. We do not advocate simply tuning parameters to get the "right" result, but rather increasing the physiological parameter space constrained by laboratory and observational work in order to create a more robust and representative model of the phytoplankton community.

*Author contributions.*   S. Clayton, S. Dutkiewicz and M. J. Follows designed the study. S. Dutkiewicz, O. Jahn, P. Heimbach and C. Hill

implemented and ran the model simulations. S. Clayton analyzed the model outputs. S. Clayton prepared the manuscript with contributions from all co-authors. The authors declare that they have no conflict of interest.

*Acknowledgements.*   We thank Dennis McGillicuddy and Amala Mahadevan for helpful discussions. This study was funded by grants from the National Science Foundation, the Gordon and Betty Moore Foundation, and the Simmons Collaboration on Ocean Processes and Ecology. SC was funded by a Moore/Sloan Data Science and Washington Research Foundation Innovation in Data Science post-doctoral fellowship

at the University of Washington. PH was supported through NASA's Physical Oceanography program.





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





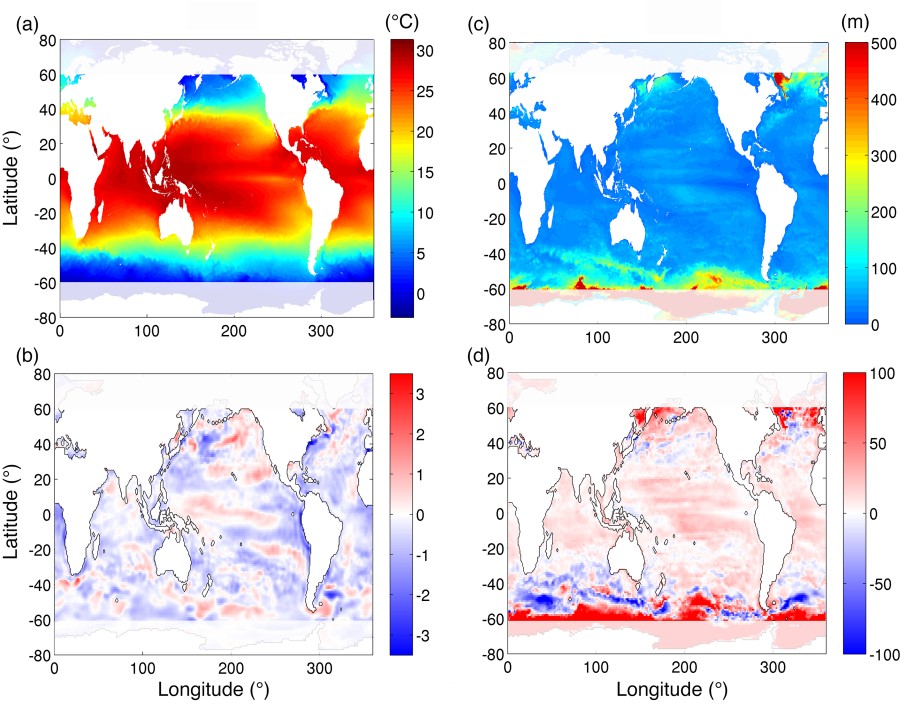

**Figure 1.** (a) Annual average sea surface temperature (SST), and (c) annual average mixed layer depths (MLD) in the HR simulation for the 1999 model year; (b) the difference in SST, and (d) in MLD between the two simulations. Positive values indicate higher values in the HR simulation and negative values indicate higher values in the CR simulation.





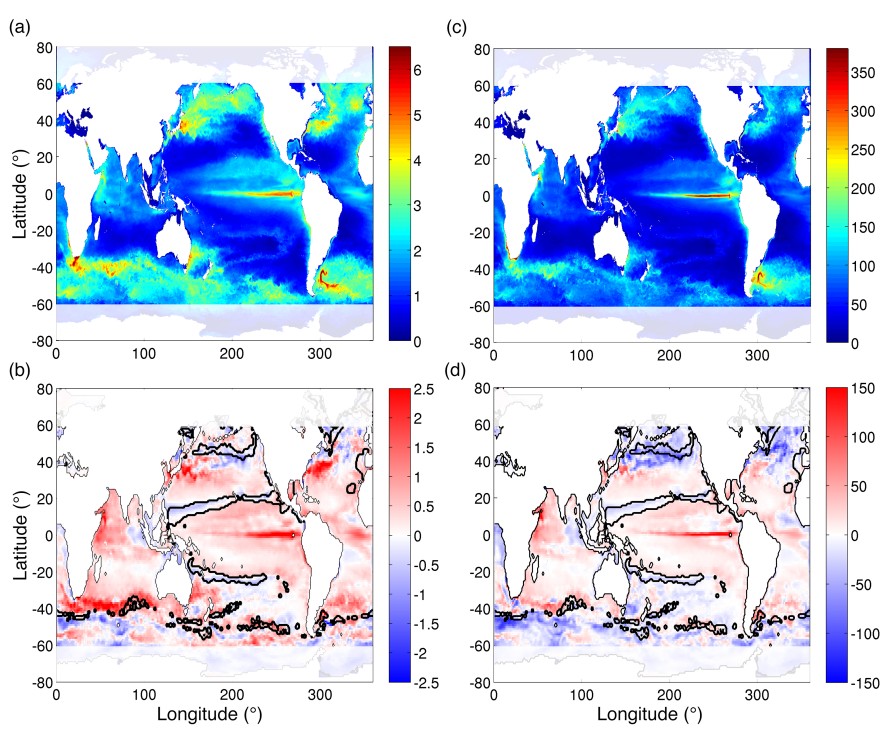

**Figure 2.** (a) Annual average total phytoplankton biomass in g C m$^{-2}$, and (c) annual primary production in g C m$^{-2}$ in the HR model solution for 1999. (b) the difference in phytoplankton biomass, and (d) the difference in annual primary production. Positive values indicate higher values in the HR simulation and negative values indicate higher values in the CR simulation. The solid black contour lines in (b) and (d) indicate the region where the limiting nutrient differs between the two model simulations





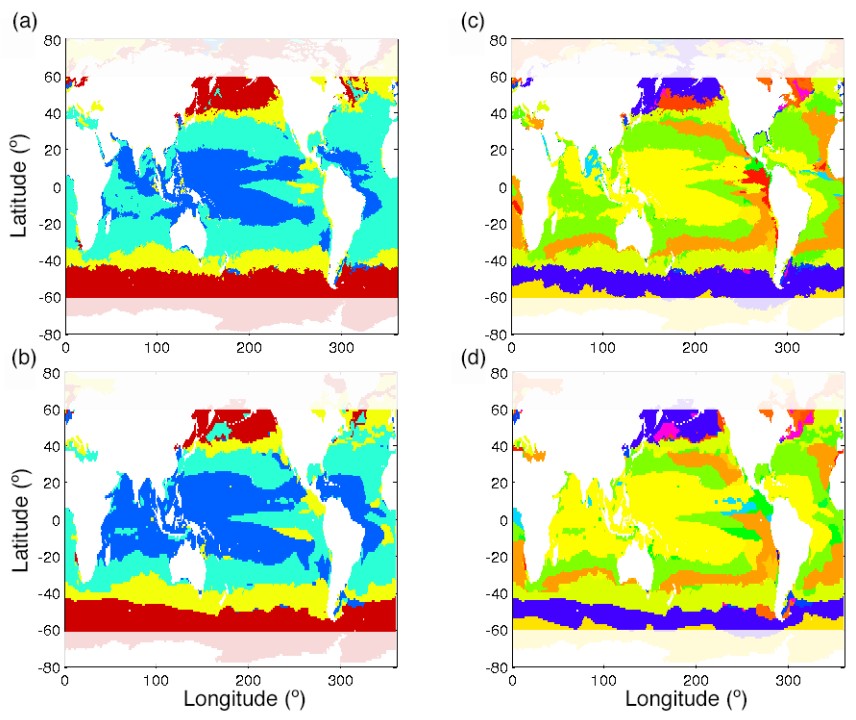

**Figure 3.** Dominant phytoplankton functional group (a) in the HR simulation, and (b) in the CR simulation. Diatoms are shown in red, large phytoplankton in yellow, picophytoplankton in green and *Prochlorococcus*-like phenotypes in blue. Dominant phytoplankton phenotype (c) in the HR simulation, and (d) in the CR simulation. Each color represents a different phenotype.





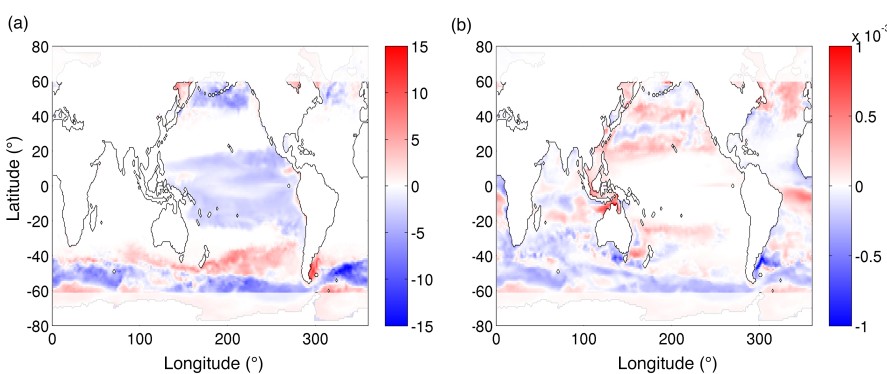

**Figure 4.** (a) Difference in the annual average surface concentrations of nitrate (mmol N m$^{-3}$) and, (b) dissolved iron (mmol Fe m$^{-3}$). Positive values indicate higher values in the HR simulation and negative values indicate higher values in the CR simulation.





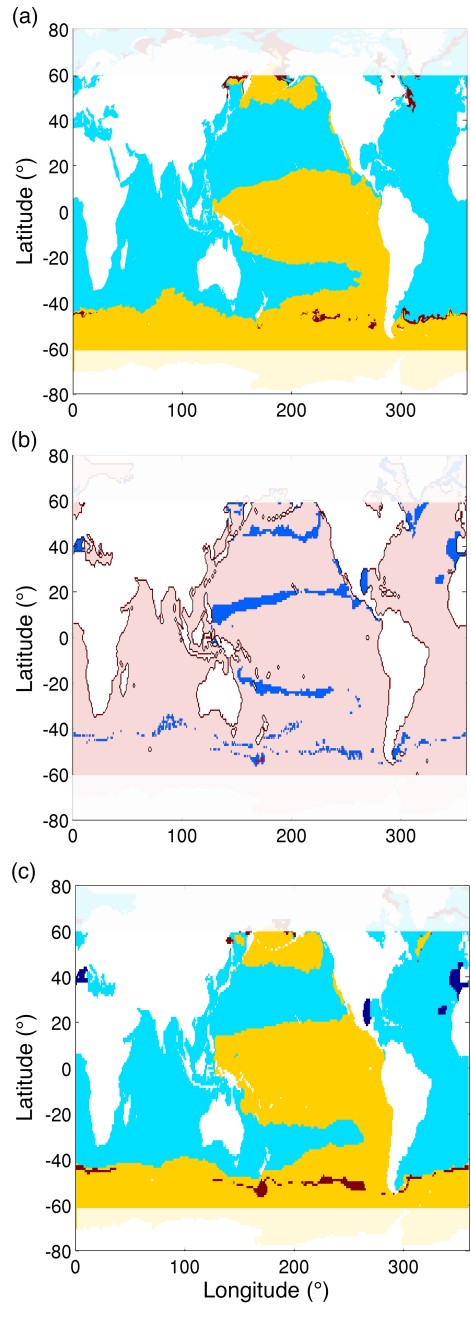

**Figure 5.** Limiting nutrients were assessed based on a biomass weighted average of the most limiting nutrient for each of the 78 total phytoplankton types (a) in the HR model, and (c) in the CR model. Iron-limited regions are shown in orange, nitrate-limited in light blue, phosphate-limited in dark blue, and silica-limited in red. (b) Regions where the model simulations predict different limiting nutrients are shown in dark blue, whereas pink regions have in the same limiting nutrient in both simulations.





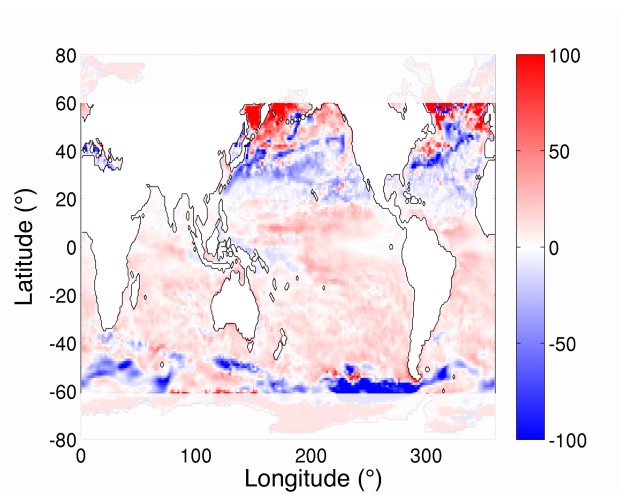

**Figure 6.** Difference in the mean mixed layer depths for March, April, May (m). Positive values indicate higher values in the HR simulation and negative values indicate higher values in the CR simulation.





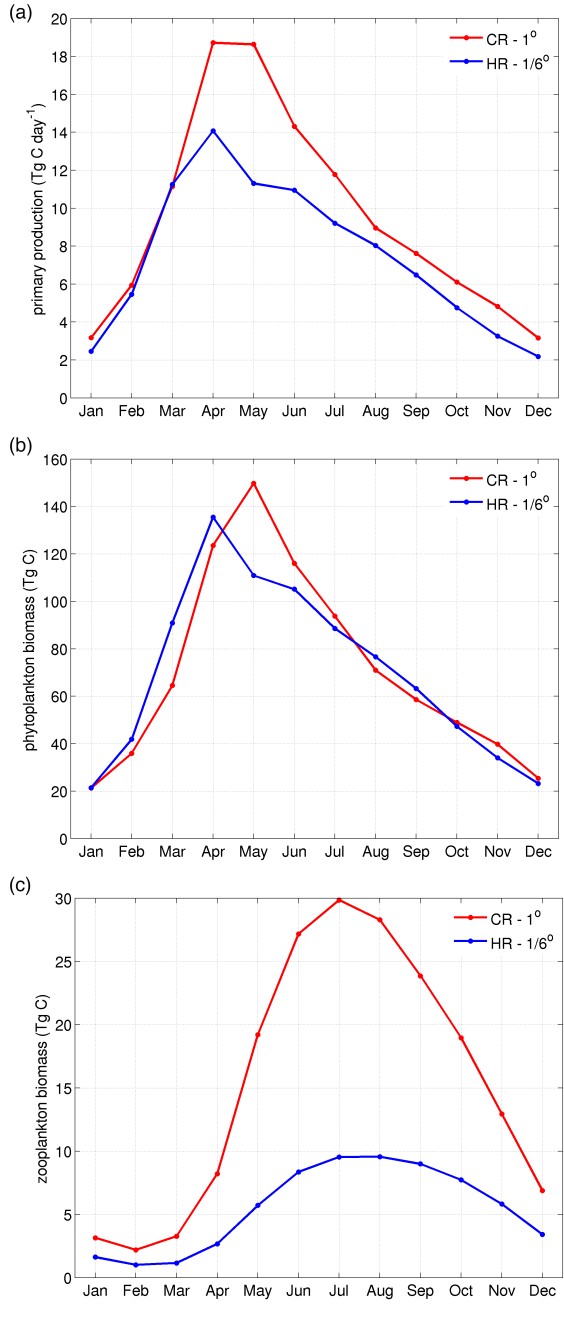

**Figure 7.** Total monthly integrated (a) primary production, (b) phytoplankton biomass and (c) zooplankton biomass for northern hemisphere ocean regions north of $40^{o}$N.