# Peer review of "Biogeochemical versus ecological consequences of modeled ocean physics"

_Biogeosciences, 2016_

## Referee Comment (RC1) · Anonymous Referee #1 · 23 Oct 2016

Note: please refer to the pdf attachment, as equations/formatting did not all paste correctly here.

Summary Statement: Clayton and colleagues explore the importance of model resolution on ocean ecology and biogeochemistry in two versions of ECCO. Both models were coupled to the same version of the Darwin ecosystem/biogeochemistry model with the primary difference between the two being 1deg verus 1/6 deg resolution, with the later considered 'eddy permitting.' The primary finding is that the phytoplankton biogeography in terms of functional type was much more stable than rates and stocks, such as primary productivity and total phytoplankton and zooplankton abundance. These conclusions are somewhat unexpected and a valuable contribution to understanding how mesoscale physics may interact with ecosystem and biogeochemical dynamics.

[Figure]

The manuscript addresses an important topic relevant to Biogeosciences and reaches novel conclusions. It is well written and structured. With some revision, I feel the manuscript will be a valuable contribution to Biogeosciences.

I do have some concerns regarding how the authors interpret differences between HR and CR simulations at the regional scale, particularly for the in the northern high latitudes. These concerns as well as minor comments are listed below.

General Comments: P8-L12-32: Interpretation is primarily through a steady-state framework (i.e., R*, P*,Z*). This holds when the timescale of physics is » timescale of biology, supply is constant, etc. My largest criticism is the attempt of the authors to explain the "Dilution-Recoupling Hypothesis" using a steady-state framework (Page 7, line 31 – Page 8 line 1). From Fig 7, it is clear that dP/dt and dZ/dt are almost never close to zero, and certainly not simultaneously zero. The hypothesis is fundamentally driven by perturbed systems where steady-state is not valid. I suggest either further justification why the authors think the Z*/P* framework is applicable here, or to use non-steady-state arguments.

âǎć Would not the deeper spring ML depth in HR simulations result in a greater annual SN? The authors argue that higher Z* in CR is due to higher SN, but present no evidence that SN is higher in CR. If anything, it seems SN should be higher for HR?

âǎć Comparing Fig 7a and 7b along with the relationship PProd = mu*P, it seems that P growth rate is higher year-round in the CR simulation. Light limitation could explain the difference in spring, as the authors point out, but what about the rest of the year?

Equatorial upwelling is minimally addressed. In Fig 2 there is a dramatic increase in phytoplankton stock and primary productivity in the Equatorial Pacific. It would be interesting to diagnose if there is a change in net nutrient supply from equatorial upwelling and if there is a change in subsequent meridional transport of nutrients, or if they are effectively locally trapped. The authors mention a change in equatorial productivity related to a change in poleward Ekman nutrient transport, but there is no discussion if

there is any change in the supply rate from upwelling (rather than mld changes). Could the authors compare over an equatorial band between the two model resolutions?

Another general comment is that the model includes two Zooplankton that I believe have size specific grazing preference. It would be interesting to diagnose if the change in resolution causes any systematic changes in the efficiency of predator-prey coupling. i.e., is there any change in the average 'g' term? One might expect that higher resolution physics could disrupt predator-prey coupling. Have the authors looked into this? (This is just my curiosity, and I am fine if the authors feel the topic is beyond the scope of this manuscript).

P8-L2-3: In equations (5) and (6) it is unclear to me why there is an 'R' term in the right side of each equation. The description of an idealized light-limited model system implies replete nutrients. Under nutrient replete conditions (R » k), the R/(R+k) term in Eq (1) approaches 1, and eq 5 should simplify to:

I do not see the need or justification for . In fact, there is no change in solutions for N* and Z* (Eqs. (9) and (10)). Equation (8) would then be superfluous and should be removed.

Minor comments: P8 L30-32: 1/6deg definitely doesn't represent the Mahadevan (2012) restratification mechanism which is fundamentally submesoscale. It might capture the McGillicuddy (2003) mechanism.

Figure captions: General comment: Indicate in each figure caption if results are annual averages for 1999, or some other time period. Figure 3: The 'green' in (a) and (b) looks quite blue/teal to me Figure 5: I would suggest that the order of (b) and (c) be switched. Also, in panel (b), it would be useful to color code by what the There are also some distracting red dots, such as south of New Zealand, that appear to be islands.

Figure 6: Does 'higher' mean deeper? Suggest that you say Positive values indicate deeper...

Figure 7: It would help interpretation to also include the seasonality of mixed layer depth for HR and CR as a separate panel (or overlay on the existing panels). Consider including seasonality of surface nitrate also.

Editorial Comments: P5-L15: tus thus P6-L31: m would be preferable, as alone typically refers to realized growth rate, not maximum growth rate. Also, SN SR P8-L19 "drives in an.." "drives an.." P9-L4 Northern and Southern Pacific North and South Pacific Fig 2 caption: Although 'annual' is in the text, units for primary production of g C m-2 y-1 would be preferable such that the units are consistent with a rate.

Please also note the supplement to this comment:
http://www.biogeosciences-discuss.net/bg-2016-337/bg-2016-337-RC1-supplement.pdf

---

## Referee Comment (RC2) · Anonymous Referee #2 · 28 Nov 2016

This study compares the simulated biological differences in a "diverse ecosystem" model coupled to two different global physical model configurations, one that has a relatively high resolution and is eddy permitting and one that is much coarser and does not resolve eddies. This is an interesting starting point and the abstract of this manuscript is quite promising. I was very curious to read about the insights that might be gained from the study laid out there. However, the manuscript itself was rather superficial and disappointing and did not deliver on the study's potential. There are some similarities and some difference between the two configuration, as one might have expected. The authors do not dig deep enough in explaining the underlying reasons for the differences. In the discussion there are several occurrences of "we hypothesise that these differences result from. . ." which is rather unsatisfying in a modelling study where one can examine every process in great detail and get to the bottom of

differences. Furthermore, the mere description of differences between the two model configurations without any comparison with observations seems rather limited. Most importantly, I don't find any broadly applicable insights articulated in the manuscript. What is the novel insight that the authors are trying to present here? Given the distinguished author list, this was a particularly disappointing read.

Some specific comments:

Intro, 1st paragraph: Higher ecological complexity is not necessarily because of the potential problem of overfitting. It would be appropriate to at least mention this.

Page 2, line 25: Reference to LeQuere et al. (2005) seems inappropriate here. This paper is not describing a global biogeochemical model, but merely a plan or idea of such.

Section 2.1: What type of data assimilation was applied to the models and could this affect the results? It has been shown previously that data-assimilative physical model solutions can lead to drastically altered biogeochemical results compared to their corresponding non-assimilative model versions (see, e.g. Raghukumar et al. Progress in Oceanography, 2015).

Figures 1 and 2: I would prefer to also see the CR results, not just HR and the differences between HR and CR.

Page 4, line 17-18 and Page 5, line 13-14: Differences between both models (model physics as well as biogeochemistry) have previously been described by Clayton et al. (2013). I'm wondering what the new and distinct contribution of this publication is in comparison to Clayton et al. (2013).

---

## Referee Comment (RC3) · Anonymous Referee #3 · 2 Dec 2016

In this study the authors investigate how model resolution influences simulated ecosystem and surface ocean biogeochemical properties. Two physical ocean configurations are used; a high-resolution eddy permitting model and a lower resolution version of the model that does not resolve eddies. When the same "emergent" biogeochemical ecosystem model is coupled to these different physical configurations the authors find that phytoplankton biogeography is similar, while other biogeochemical properties have much larger differences. Investigating the "biogeochemical versus ecological consequences of modeled ocean physics" is important and I was eager to read such a study. However, I was somewhat disappointed with what was presented. The writing is clear and the analysis to show how the model results differ is mostly acceptable. However, I was not satisfied with the explanation of why there are similarities and differences between the two set-ups. The authors did not conduct a deep enough

investigation and it seemed more that they were simply showing similarities and differences and then hypothesizing why this occurred. I realize that the authors made some attempts to figure out the reasons behind the similarities and differences using a resource competition framework, but they did not take this investigation far enough and instead often ended up concluding their investigation by saying things like, "we hypothesize..." or "this may...". This is rather unsatisfying since one should be able to examine the model results in detail to actually determine why any similarities or differences occurred. Moreover, after reading Clayton et al., 2013 again, it seemed to me as if the authors are merely trying to extend their earlier work and publish a few new details that probably should have just been included in the earlier publication (i.e., not much seems to be new except for a few plots of biogeochemical differences). Am I wrong in this or is this a new set of experiments? The methods section of the paper was also lacking a few details and I had to assume that the set-up was the same as in the earlier paper based on what was stated in the results and discussion section. Without some of the critical information on how the model was spun-up and more crucially, for how long it was spun-up, I also had a difficult time interpreting some of the presented results. If the model was only run for 8 years as in Clayton et al., 2013 then, I highly doubt that steady state or even quasi-steady state conditions were reached. This makes it challenging to investigate biogeochemical properties because of model 'drift'. While it may be possible to somewhat account for such 'drift' the authors have not attempted to do so and thus, have only provided a snapshot of a system that would likely be quite different if the simulations were run for a longer period of time. I realize that there are computational limitations that prevent high-resolution models from easily being run to steady state, but the authors need to address the issue of 'drift' if they want to investigate differences in biogeochemistry. This is an issue even in an idealized case where the goal is not to reproduce observations, but to only compare differences due to model resolution. Overall, I also found myself wondering what the important insights from the study were. Yes, the message is that there are some similarities and differences that could be important, but what does it mean for the marine biogeochemical

modelling community? The few concluding statements are not very satisfying.

Specific comments:

As mentioned above, some critical information is missing from the Methods section. Information on how long the model was spun-up for is needed. More information is also needed on the biogeochemical forcing data. What biogeochemical data sets are used to initialize the model? Is this World Ocean Atlas data, etc.?

An analysis should be conducted to address the issue of model 'drift', i.e., how much drift is occurring and what it might mean for interpreting the results. The resource competition framework that is used to explain some of the differences depends on the system being at steady state to work. If this is not the case as I suspect then it's difficult to see how such a framework can be used to explain the differences. The authors will need to provide more evidence for this to be believable.

Is annual averaging the best way to evaluate the similarities and differences that are seen in Figs. 1-5? As Figure 7 shows there are striking monthly differences at higher latitudes. Perhaps it would be more informative to compare and show the key physical, ecological, biogeochemical properties in seasonal plots (e.g., winter, summer, fall, and spring)? Or maybe carefully selected Hövemoller type plots would be informative? I would be particularly interested in seeing if phytoplankton diversity and differences are more pronounced seasonally or during the progression of the spring bloom in the Atlantic.

In Figs. 1 and 2, it would be nice to see the CR results too.

Fig. 3. I find that this lone figure made it difficult to really see the differences between the model configurations and found that I had to refer to Clayton et al., 2013 to really understand what was going on. This is somewhat frustrating, as some of this information seems to be necessary to understand the study. It would be really nice to have this all in one publication.

Fig. 4. What is the actual concentration? Is it realistic? I realize that the purpose of the study is not to figure out which is the 'best' simulation, but it would still be nice to see the absolute values.

Page 5 line 15 'tus' needs to be 'thus'

---

## Author Comment (AC1) · 1 Mar 2017

**bg-2016-337**
**Biogeochemical versus ecological consequences of modeled ocean physics**
Sophie Clayton, Stephanie Dutkiewicz, Oliver Jahn, Christopher Hill, Patrick Heimbach, and Michael J. Follows

**Response to reviewers:**

We thank the three reviewers for their helpful comments on the manuscript. We have set out our responses to the reviewers' comments below, in blue. As instructed, we have not prepared a revised version of the manuscript, but we have included detailed notes of changes that will be incorporated into the revised manuscript.

We would like to address some of the same concerns raised by Reviewers #2 and #3:

Firstly, both reviewers were unclear on what the main message of the paper is. In this study, we examined the different responses of a diverse ecosystem model to a coarse resolution and an eddying general circulation model. We found clear and unexpected differences, namely that although there were big differences in the bulk biogeochemical properties of the model solutions, the realized phytoplankton ecosystem had essentially the same structure. As pointed out by Reviewer #1, this is an interesting and unexpected result. The clear message of this paper is that the apparent decoupling in the biogeochemical and ecological responses of the model to different physical forcing can be well explained and is not a random response to the different forcing. In the discussion section of this paper, we have highlighted the mechanisms responsible for the apparent disconnected between the biogeochemical and ecological responses to the different modeled ocean physics. We should also state that this is, to our knowledge, the only study that examines the different responses of a complex ecosystem model coupled to global ocean circulation models at differing resolution. We will make these points more strongly in the revised version of the manuscript as it seems that some of the reviewers were not clear on these points.

Secondly, we would like to respond to the point raised by both Reviewers #2 and #3, that they believe that this manuscript is not sufficiently different from Clayton et al (2013) to warrant publication as a separate study. We respectfully, but very strongly disagree with their assessment of the work under discussion here. This work is based on the same set of model experiments, but that is all that it shares. This work is a stand-alone study from Clayton et al (2013). The previous paper was entirely focused on understanding the underlying physical controls (primarily transport) on modeled patterns of phytoplankton biodiversity. In fact the different resolution of the models was only a very small portion of that paper. The earlier paper does not consider any of the differences in the other ecological properties of the model results (e.g. dominant phenotype or functional group) or in the biogeochemical properties of the model. Specifically, we would draw the reviewers' and the editor's attention to the fact that there is not a single figure shared between these two manuscripts.

Thirdly, Reviewers #2 and #3 commented on what they see as a prevalence of woolly statements in the manuscript, specifically the use of the phrases "we hypothesize that…" and "may be…". We point out that we have used the phrase "we hypothesize that…" exactly twice in this manuscript (P7, L31-32; P8, L23), and that both of these instances

are found in the same section (4.2). We have used the phrase "this may…" only once to speculate on the drivers of model differences (P7, L17), at the end of section 4.1. We respectfully but strongly disagree with the reviewers: our arguments in section 4.1, 4.3 and 4.4. are strong and we have spent considerable care in laying these out. We have fully explained the mechanisms driving differences in the system in the subtropical gyres and at the gyre boundaries. We however do recognize that section 4.2 could be improved, so we have set out below (primarily in response to Reviewer #1's comments) our improvements to that section in the revised manuscript.

Please find our responses to specific points below.

**Reviewer #1**

Clayton and colleagues explore the importance of model resolution on ocean ecology and biogeochemistry in two versions of ECCO. Both models were coupled to the same version of the Darwin ecosystem/biogeochemistry model with the primary difference between the two being 1deg versus 1/6 deg resolution, with the later considered 'eddy permitting.' The primary finding is that the phytoplankton biogeography in terms of functional type was much more stable than rates and stocks, such as primary productivity and total phytoplankton and zooplankton abundance. These conclusions are somewhat unexpected and a valuable contribution to understanding how mesoscale physics may interact with ecosystem and biogeochemical dynamics.

The manuscript addresses an important topic relevant to Biogeosciences and reaches novel conclusions. It is well written and structured. With some revision, I feel the manuscript will be a valuable contribution to Biogeosciences.

I do have some concerns regarding how the authors interpret differences between HR and CR simulations at the regional scale, particularly for the in the northern high latitudes. These concerns as well as minor comments are listed below.

General Comments:
P8-L12-32: Interpretation is primarily through a steady-state framework (i.e., $R^*$, $P^*$, $Z^*$). This holds when the timescale of physics is >> timescale of biology, supply is constant, etc. My largest criticism is the attempt of the authors to explain the "Dilution-Recoupling Hypothesis" using a steady-state framework (Page 7, line 31 – Page 8 line 1). From Fig 7, it is clear that dP/dt and dZ/dt are almost never close to zero, and certainly not simultaneously zero. The hypothesis is fundamentally driven by perturbed systems where steady-state is not valid. I suggest either further justification why the authors think the $Z^*$/$P^*$ framework is applicable here, or to use non-steady-state arguments.

Thank you for these valuable comments. We recognize that in this context, the dilution-recoupling hypothesis was not appropriate, and we have removed the reference to it in the revised manuscript.

Here, we invoke the steady state framework as way of explaining the apparent disconnect between the similar phytoplankton biomass abundance, and simultaneously very different zooplankton abundances between simulations (in the subpolar gyres). This signal is overwhelmingly driven by the behavior of the system during the summer and

autumn months, after the spring bloom. During this period, we do believe that a steady-state approximation is appropriate, as the MLDs and ecosystem properties are stable over several months (see Fig. R1, below). However, we do agree that this is not appropriate when considering either the winter months, or the entire annual cycle.

• Would not the deeper spring ML depth in HR simulations result in a greater annual $S_N$? The authors argue that higher Z* in CR is due to higher $S_N$, but present no evidence that $S_N$ is higher in CR. If anything, it seems $S_N$ should be higher for HR?
      We have evaluated the annual mean $wNO_3$ at 100m, analogous to $S_N$. In the high latitudes, $wNO_3$ is 142.1 mmol $NO_3$ $m^{-2}$ $year^{-1}$ in CR, and 108.1 mmol $NO_3$ $m^{-2}$ $year^{-1}$ in HR. We will include these results in the revised version of section 4.2 (which we have included below). So $S_N$ is higher in CR than HR in the high latitudes, despite the deeper MLDs in HR. This difference is due to the higher variability in vertical velocities in HR, and the asymmetry in vertical transport of nutrients. Nutrient concentrations in the surface mixed layer are lower than those below it, so higher variability in vertical velocities will result in decreased nutrient supply compared to the less variable case.

• Comparing Fig 7a and 7b along with the relationship PProd = mu*P, it seems that P growth rate is higher year-round in the CR simulation. Light limitation could explain the difference in spring, as the authors point out, but what about the rest of the year?
      We have replaced Fig. 7 with a Hovmoller diagram (Fig. R1, below), which better represents seasonality. The previous version of the figure confounded seasonal and latitudinal differences between the simulations.

      In the context of this model, the limiting factors on the phytoplankton growth rate are multiplicative, e.g. for a modeled phytoplankton phenotype *j*, the growth term is given by:

$$\mu_j = \mu_{MAX}\, \gamma_j^T \gamma_j^I\, \frac{R}{R + k_{Rj}}$$

where $\gamma^T$ and $\gamma^I$ are the temperature and light limitation terms, respectively.

In the Northern Hemisphere:
During the winter, when nutrients are replete (and the nutrient term goes to 1), if either the light or temperature fields experienced by the modeled phytoplankton are consistently less favourable, then $\mu$ will be lower. MLDs are consistently deeper in the HR simulation during the winter months, resulting in lower winter PP in HR than CR.

The onset of the spring bloom occurs roughly one month earlier (March-April) in the HR simulation than the CR simulation (April-May). This can be seen as higher PP in the HR simulation in March and April, followed by a reversal with higher PP in the CR simulation in May. This is driven by differences in the timing of the shoaling of the MLDs.

Through the summer and into autumn, the MLD shoals and nutrients become limiting. Summer MLDs are deeper and there is a higher $wNO_3$ in CR, resulting in higher $S_R$ and surface nitrate concentrations in CR than HR over the summer months. This explains the higher PP during early summer in the CR simulation.

Similar patterns can be seen in the Southern Hemisphere, where the MLD shoals earlier in the spring in the HR simulation, and PP and P biomass are both lower in the HR simulation during the summer.

Equatorial upwelling is minimally addressed. In Fig 2 there is a dramatic increase in phytoplankton stock and primary productivity in the Equatorial Pacific. It would be interesting to diagnose if there is a change in net nutrient supply from equatorial upwelling and if there is a change in subsequent meridional transport of nutrients, or if they are effectively locally trapped. The authors mention a change in equatorial productivity related to a change in poleward Ekman nutrient transport, but there is no discussion if there is any change in the supply rate from upwelling (rather than mld changes). Could the authors compare <wN> over an equatorial band between the two model resolutions?

Thank you for suggesting that we look at wN. We have evaluated the annual average $wNO_3$ at 100m between 5°N and 5°S for the both models. The regionally integrated mean annual vertical $NO_3$ fluxes (mmol $NO_3$ $m^{-2}$ $year^{-1}$) evaluated at 100m for model year 1999 were 453.1 mmol $NO_3$ $m^{-2}$ $year^{-1}$ and 383.7 mmol $NO_3$ $m^{-2}$ $year^{-1}$, for the HR and CR simulations, respectively. There is a clear increase in vertical nutrient supply in the equatorial upwelling zone in HR that can account for the dramatic increase in phytoplankton stock and primary productivity in the Equatorial Pacific.

Another general comment is that the model includes two Zooplankton that I believe have size specific grazing preference. It would be interesting to diagnose if the change in resolution causes any systematic changes in the efficiency of predator-prey coupling. i.e., is there any change in the average 'g' term? One might expect that higher resolution physics could disrupt predator-prey coupling. Have the authors looked into this? (This is just my curiosity, and I am fine if the authors feel the topic is beyond the scope of this manuscript).

Yes, the ecological model includes two zooplankton. Small and large zooplankton graze preferentially on small and large phytoplankton, respectively. Although this is an interesting question, we feel that it is beyond the scope of this manuscript, as we would need to re-run both models in order to diagnose the 'g' term to explore differences in it.

P8-L2-3: In equations (5) and (6) it is unclear to me why there is an 'R' term in the right side of each equation. The description of an idealized light-limited model system implies replete nutrients. Under nutrient replete conditions (R >> k), the R/(R+k) term in Eq (1) approaches 1, and eq 5 should simplify to:
dR/dt = - $\mu_l$P + $S_R$ + $\gamma m_Z$Z
I do not see the need or justification for $\mu_l$RP. In fact, there is no change in solutions for N* and Z* (Eqs. (9) and (10)). Equation (8) would then be superfluous and should be removed.

This is correct. However, we have revised our analysis in this section (4.2), and find that nutrient limitation is a more appropriate way to explain the behaviour of the system during the summer. We have updated our equations accordingly to the following:

$$\frac{dR}{dt} = -\mu_{max}\frac{RP}{R+k} + S_R$$

$$\frac{dP}{dt} = \mu_{max} \frac{RP}{R + k} - gZP$$

$$\frac{dZ}{dt} = gZP - m_z Z$$

with the following solutions for a steady system:

$$R^* = \frac{kS_R\, g}{-gS_R + \mu_{max} m_z}$$

$$P^* = \frac{m_z}{g}$$

$$Z^* = k(\frac{S_R\, g}{\mu_{max} m_z} - 1)$$

The increased $S_R$ in the CR simulation (due to higher wN and deeper summer MLD) can account for the increased zooplankton abundance in CR in the summertime, although the summer (June - October) phytoplankton concentration remains very similar between both simulations.

Minor comments:
P8 L30-32: 1/6deg definitely doesn't represent the Mahadevan (2012) restratification mechanism which is fundamentally submesoscale. It might capture the McGillicuddy (2003) mechanism.
         You are correct, the Mahadevan (2012) restratification mechanism is not resolved in the HR model. We had initially included that reference to give broader context on the known effects of fine sale physical dynamics, however it seems that it is superfluous in the context of this model study, so we will remove that sentence from the revised manuscript.

Figure captions:
General comment: Indicate in each figure caption if results are annual averages for 1999, or some other time period.
         We will add this to each of the figure captions where it is missing in the revised manuscript.

Figure 3: The 'green' in (a) and (b) looks quite blue/teal to me
         Could this be an issue with your monitor? The green colour in Fig. 3 looks green on my monitor and in my printed out version of the manuscript.

Figure 5: I would suggest that the order of (b) and (c) be switched. Also, in panel (b), it would be useful to color code by what the
         Something seems to be missing in the above comment.
There are also some distracting red dots, such as south of New Zealand, that appear to be islands.
         Yes, the red dots are islands, we will update Fig. 5 in the revised manuscript to remove them.

Figure 6: Does 'higher' mean deeper? Suggest that you say Positive values indicate deeper…

> Yes, thank you for pointing that out, it is a little confusing. We will change the figure caption for Fig. 1 to read "Positive values indicate deeper MLD in the HR simulation, and negative values indicate deeper MLD in the CR simulation". We have removed Fig. 6 from the revised manuscript.

Figure 7: It would help interpretation to also include the seasonality of mixed layer depth for HR and CR as a separate panel (or overlay on the existing panels). Consider including seasonality of surface nitrate also.

> We have replaced Figs. 6 and 7 with a Hovmoller diagram (Fig. R1, below) which now includes the seasonality of the MLD and surface nitrate.

Editorial Comments:

P5-L15: tus to thus

> We will make the correction in the revised manuscript.

P6-L31: µ to $\mu_m$ would be preferable, as µ alone typically refers to realized growth rate, not maximum growth rate. Also, $S_N$ to $S_R$

> We have modified eqs 1 and 2 to the following, and make the changes consistent through the rest of section 4.1

$$\frac{dR}{dt} = -\mu_{max}\frac{RP}{R+k} + S_R$$

$$\frac{dP}{dt} = \mu_{max}\frac{RP}{R+k} - mP$$

P8-L19 "drives in an.." to "drives an.."

> We will correct that in the revised manuscript.

P9-L4 Northern and Southern Pacific to North and South Pacific

> We will make those changes in the revised manuscript.

Fig 2 caption: Although 'annual' is in the text, units for primary production of g C m$^{-2}$ y$^{-1}$ would be preferable such that the units are consistent with a rate.

> We will make that change in the caption for Fig. 2 in the revised manuscript.

**Reviewer #2**

This study compares the simulated biological differences in a "diverse ecosystem" model coupled to two different global physical model configurations, one that has a relatively high resolution and is eddy permitting and one that is much coarser and does not resolve eddies. This is an interesting starting point and the abstract of this manuscript is quite promising. I was very curious to read about the insights that might be gained from the study laid out there. However, the manuscript itself was rather superficial and disappointing and did not deliver on the study's potential.

There are some similarities and some difference between the two configuration, as one might have expected.

That there are differences and similarities is possibly expected, but the fact that the ecological solution of the model (phytoplankton community structure) is not affected by changes in the model physics, whereas the modeled biogeochemical properties of the model are greatly affected by changes in the model physics is striking and unexpected (see Reviewer #1's comments).

The authors do not dig deep enough in explaining the underlying reasons for the differences. In the discussion there are several occurrences of "we hypothesise that these differences result from. . ." which is rather unsatisfying in a modeling study where one can examine every process in great detail and get to the bottom of differences.

We respectfully disagree with the reviewer that we do not explain the underlying reasons. We point out that we make very definitive statements in section 4.1 about the processes that drive the differences between models in the subtropical gyres, as well as in section 4.3 about the processes that account for the geographical shifts in the boundaries between biogeographical provinces. We note that the phrase "we hypothesize that…" occurs exactly twice in the submitted manuscript (P7, L31-32; P8, L23), with both instances found in the same section (4.2). We do not believe that this can be described as "several occurrences".

However we do recognize that our discussion of the processes driving the changes in the northern hemisphere high latitudes in section 4.2 may have been a little less concrete in the submitted manuscript and we have strengthened it in with the addition of more analysis, as set out in our responses to the specific reviewer comments.

Furthermore, the mere description of differences between the two model configurations without any comparison with observations seems rather limited.

As stated in the introduction of our manuscript, the intent of this study was not to assess which model performs best with respect to reality, but rather to better understand how differences in the modeled ocean physics might affect *both* the ecological and biogeochemical properties of the model solutions. We have used versions of the CR model in several previous studies where we have compared model output more thoroughly to observations (see e.g. Follows et al 2007; Dutkiewicz et al., 2012; 2015). We feel that additional comparisons to observations would detract from the main message of the work, which explains how, in different biogeographical regions, the modeled phytoplankton community remains unchanged whereas the biogeochemical properties of the model, which are ultimately set by the phytoplankton community, vary greatly with changes in the modeled ocean physics. We also disagree that the manuscript is merely a "description" of differences. The main point of the paper is to explain the differences and similarities.

Most importantly, I don't find any broadly applicable insights articulated in the manuscript. What is the novel insight that the authors are trying to present here? Given the distinguished author list, this was a particularly disappointing read.

We are disturbed that the reviewer found no "insights articulated" in this study. We believe that we have shown that the bulk biogeochemical properties of this ecological model (*which has many similarities to other widely used biogeochemical models which also resolve multiple phytoplankton PFTs*) are more sensitive to

differences in modeled ocean physics than the structure of the ecosystem itself. We would argue that this actually has profound implications for how we might think about structuring ecological and biogeochemical models. We have made this point more clearly in the conclusions of the revised version of the manuscript as it seems that it was not entirely clear to this reviewer. We will add an additional paragraph to the conclusions:

> "We have shown that the bulk biogeochemical properties of this ecological model are more sensitive to differences in modeled ocean physics than the structure of the ecosystem itself. Given that this model has many similarities to other widely used biogeochemical models, which also resolve multiple phytoplankton PFTs, this study provides important insights into how these models might behave under different physical conditions."

Some specific comments:
Intro, 1st paragraph: Higher ecological complexity is not necessarily because of the potential problem of overfitting. It would be appropriate to at least mention this.
    We are not entirely sure what the reviewer is referring to here. As we are not constraining any of the biogeochemical or ecological fields to observations, it is hard to see how adding biological complexity could result in overfitting. Possibly the reviewer is mistaken in thinking that the biogeochemistry or ecosystem in this study is assimilated?

    As the goal is to adequately represent the system under consideration, surely the level of complexity needed to represent the system will vary depending on the question being posed? We do not feel that any of our statements in the first paragraph of the introduction contradict this point.

Page 2, line 25: Reference to LeQuere et al. (2005) seems inappropriate here. This paper is not describing a global biogeochemical model, but merely a plan or idea of such.
    We disagree on this point. Le Quéré et al (2005) explicitly discusses how biogeochemical models that resolve phytoplankton functional types (PFTs) behave differently to simple NPZD models. For example, their Fig. 8 shows the differences in modeled chlorophyll *a* for an NPZD model versus the PISCES model (with 3 PFTs) and the Dynamic Green Ocean Model (with 4 PFTs). We believe that this is a perfectly relevant reference in this context.

Section 2.1: What type of data assimilation was applied to the models and could this affect the results? It has been shown previously that data-assimilative physical model solutions can lead to drastically altered biogeochemical results compared to their corresponding non-assimilative model versions (see, e.g. Raghukumar et al. Progress in Oceanography, 2015).
    Both physical models are data assimilation products. The ECCO-GODAE product is based on the Lagrange Multiplier method and the ECCO2 product employs a simplified Green's function method. We will add more detail on this in a revised version. However, there was no assimilation of biogeochemical or ecosystem variables in this study. It is possible that the reviewer did not appreciate this point, so we will also make this clearer to the text.

We note (though not sure that this is applicable to Raghukumar et al, 2015) that a reason for drastic alteration of biogeochemical results in data assimilative models is the use of so-called "filtering" methods that are designed for forecasting instead of "smoother" methods that are optimal for reconstruction. Filtering methods, such as the Kalman Filter (or optimal interpolation, which is an approximate form of the KF) incur so-called "analysis increments" at regular time intervals when new observations become available. At these times, a new initial condition is generated as a weighted sum of model forecast and observation, producing an "analysis". Importantly (and not widely appreciated in the context of reconstruction), this step violates conservation of momentum, heat and salt. As a result, artificial adjustment motions are triggered, in particular in the vertical velocity field, which is crucial for vertical transport of biologically active tracers (see discussion in Wunsch and Heimbach (2013) and Stammer et al. (2016) how smoother methods can alleviate this problem). Thus we feel we should be cautious in how we attribute large changes between data-assimilative physical/biogeochemical models relative to non-assimilative models in general (again we do not know whether this is an issue in Raghukumar et al (2015) however).

We thank the reviewer for bringing the Raghukumar et al (2015) paper to our attention. We notice that in that study, they used an earlier version of the same ecological model used in this work. However, we were very disappointed to see that they did not present any of the ecosystem level differences for the model solutions using different data assimilation methods. This is a shame, as we suspect that had they looked at differences in the phytoplankton community structure (e.g. dominant phenotypes and functional groups) their results would have echoed our findings, that although the bulk biogeochemical properties of the system are strongly modulated by different ocean physics, the phytoplankton community structure is more robust to those changes.

Although it is possible that some differences in the ECCO state estimates might be due to assimilation methods, as well as to differences in the resolved physical dynamics, the bottom line is that differences in the realized physical fields are responsible for driving different biogeochemical responses. We have added a comment to this effect in the third paragraph of the conclusions in our revised manuscript.

Figures 1 and 2: I would prefer to also see the CR results, not just HR and the differences between HR and CR.
We have opted not to show the CR results in order not to overload the paper with unnecessary and repetitive extra figures. In an earlier draft of the paper we had included

both and found that it was distracting as the basic patterns were so similar: *the difference plot is essential*. Since the focus of this paper is on the differences between the model solutions, we believe that it makes more sense to only show the results of one of the model configurations, along with the difference plot in the main text. We would be happy to add the CR panels to supplementary material in the revised manuscript.

Page 4, line 17-18 and Page 5, line 13-14: Differences between both models (model physics as well as biogeochemistry) have previously been described by Clayton et al. (2013). I'm wondering what the new and distinct contribution of this publication is in comparison to Clayton et al. (2013).
    We respectfully disagree. Clayton et al (2013) shows the difference between the following properties of the models:
  - Eddy kinetic energy (Fig. 1C)
  - SST variance (Fig. 1D)
  - Biodiversity (Fig. 3B)
  - Globally integrated annual average abundance of each modeled phenotype (Fig. 4B).

We would draw the reviewers' and the editor's attention to the fact that none of these physical or biogeochemical properties of the models are discussed (or included in figures) in the manuscript under review here. We have addressed this point above, but we would like to repeat that the manuscript under review here is a distinct contribution from the previous (2013) paper. However, we recognize that the first sentence of the results section (P4, L17) may be misleading on this point, so we have removed it from the revised manuscript and will modify the second sentence to read:

  "We describe differences in some of the physical properties of the physical circulation estimates: sea surface temperature (SST), mixed layer depth (MLD) and vertical nutrient fluxes. Although some differences in the two model configurations have previously been discussed in Clayton et al (2013), here we examine those physical processes most directly relevant to biogeochemical processes not considered in that previous work."

**Reviewer #3**

In this study the authors investigate how model resolution influences simulated ecosystem and surface ocean biogeochemical properties. Two physical ocean configurations are used; a high-resolution eddy permitting model and a lower resolution version of the model that does not resolve eddies. When the same "emergent" biogeochemical ecosystem model is coupled to these different physical configurations the authors find that phytoplankton biogeography is similar, while other biogeochemical properties have much larger differences. Investigating the "biogeochemical versus ecological consequences of modeled ocean physics" is important and I was eager to read such a study. However, I was somewhat disappointed with what was presented. The writing is clear and the analysis to show how the model results differ is mostly acceptable. However, I was not satisfied with the explanation of why there are similarities and differences between the two set-ups. The authors did not conduct a deep enough investigation and it seemed more that they were simply showing similarities and differences and then hypothesizing why this occurred. I realize that the authors made some attempts to figure out the reasons behind the similarities and differences using a

resource competition framework, but they did not take this investigation far enough and instead often ended up concluding their investigation by saying things like, "we hypothesize..." or "this may...". This is rather unsatisfying since one should be able to examine the model results in detail to actually determine why any similarities or differences occurred.

We respectively, but strongly disagree with the reviewer that we do not explain the underlying reasons for the similarities and difference. We point out that we make very definitive statements in section 4.1 about the processes that drive the differences between models in the subtropical gyres, as well as in section 4.3 about the processes which account for the geographical shifts in the boundaries between biogeographical provinces. We note that the phrase "we hypothesize that…" occurs exactly twice in the submitted manuscript (P7, L31-32; P8, L23), with both instances found in the same section (4.2). We do not feel that two occurrences of the phrase qualifies as "often". We would also mention that there is only one instance in the text where we use the phrase "this may…" to speculate on the drivers of model differences. Again, we do not feel that this qualifies as "often".

However we do recognize that our discussion of the processes driving the changes in the northern hemisphere high latitudes in section 4.2 may have been a little less concrete in the submitted manuscript and we have strengthened for our revised manuscript with the addition of more analysis, and have included it below.

Moreover, after reading Clayton et al., 2013 again, it seemed to me as if the authors are merely trying to extend their earlier work and publish a few new details that probably should have just been included in the earlier publication (i.e., not much seems to be new except for a few plots of biogeochemical differences). Am I wrong in this or is this a new set of experiments?

Again, we respectfully but strongly disagree. Yes, both papers use the same two model configurations, but the focus of the two papers is very different. This work stands alone from the previous paper, which was concerned with understanding the physical contributions, specifically with respect to transport, to modeled patterns of phytoplankton biodiversity. In fact the role of differing resolution plays only a very small part in our earlier paper. By contrast, this paper explicitly addresses the effects of model resolution on the biogeochemical and ecological model results.

The methods section of the paper was also lacking a few details and I had to assume that the set-up was the same as in the earlier paper based on what was stated in the results and discussion section. Without some of the critical information on how the model was spun-up and more crucially, for how long it was spun-up, I also had a difficult time interpreting some of the presented results. If the model was only run for 8 years as in Clayton et al., 2013 then, I highly doubt that steady state or even quasi-steady state conditions were reached. This makes it challenging to investigate biogeochemical properties because of model 'drift'. While it may be possible to somewhat account for such 'drift' the authors have not attempted to do so and thus, have only provided a snapshot of a system that would likely be quite different if the simulations were run for a longer period of time. I realize that there are computational limitations that prevent high-resolution models from easily being run to steady state, but the authors need to address the issue of 'drift' if they want to investigate differences in biogeochemistry.

Thank you for drawing our attention to the fact that the methods section would benefit from a more detailed description of the model setup, initial conditions and forcing, and the length of the simulations. We will update the revised version of the manuscript with the information outlined below.

We do not feel that model drift is an issue. Indeed it would take several thousands of years to fully spin up the biogeochemistry (in particular the deep nutrient concentrations) of the models. Even then the model would not be in steady state as seasonal cycling and interannual variability would only lead to what we might call a quasi-steady state. It is certainly not feasible to spin up the HR this long, nor do we feel that this would lead to insights into the specific issues we address in this study.

In several previous studies using the CR model (see e.g. Dutkiewicz et al., 2009; 2012; 2015) and in this study for both CR and HR we find that it takes only 3 years for the ecosystem to reach a stable annual cycle. And at this stage the drift in surface nutrients is very small – much smaller than the seasonal and interannual variability. We also looked at the trend in globally integrated annual average phytoplankton biomass, and found that it was in quasi-steady state (i.e. no identifiable trend) for the last few years of both simulations.

Both models were initialized with identical initial conditions and run for the same number of years. We are conducting a comparison of a snapshot of both of the models at the same point in time (i.e. for the model year 1999, at which point both models have been run out for 7 years from identical initial conditions). We realize that including these points in the description of the model setup would be useful to address this issue and will include a paragraph in section 2.2 addressing these issues (as outlined above) in the revised manuscript.

This is an issue even in an idealized case where the goal is not to reproduce observations, but to only compare differences due to model resolution. Overall, I also found myself wondering what the important insights from the study were. Yes, the message is that there are some similarities and differences that could be important, but what does it mean for the marine biogeochemical modelling community? The few concluding statements are not very satisfying.

We have addressed this point above and in our specific response to the same point raised by Reviewer #2.

Specific comments:

As mentioned above, some critical information is missing from the Methods section. Information on how long the model was spun-up for is needed. More information is also needed on the biogeochemical forcing data. What biogeochemical data sets are used to initialize the model? Is this World Ocean Atlas data, etc.?

Thank you for pointing this out, we agree that the methods section as indeed too terse. We will add this additional information. Both of the models were initialized with identical initial conditions, forced with identical PAR fields and aerial iron dust inputs, and run for the period from 1992-1999.

The initial conditions for both model runs were as follows:

- NO$_3$, PO$_4$, SiO$_2$ were taken from the January climatological values given in World Ocean Atlas 2005 (Garcia *et al.*, 2006)
- Each phytoplankton and zooplankton phenotype was initialized with an identical low abundance

An analysis should be conducted to address the issue of model 'drift', i.e., how much drift is occurring and what it might mean for interpreting the results. The resource competition framework that is used to explain some of the differences depends on the system being at steady state to work. If this is not the case as I suspect then it's difficult to see how such a framework can be used to explain the differences. The authors will need to provide more evidence for this to be believable.

As has been stated above, both models were initialized with identical initial conditions and run for the same length of time. As alluded to by the reviewer, it is unfeasible to run a 1/6° resolution global model with ~100 biogeochemical tracers for much longer than this, so this is a useful study for what is currently the state of the art.

We have indeed considered the issue of drift in this and in previous studies, and have found that the phytoplankton community structure is robust after about 3 years' model run time, so a run of 7-8 years is sufficient to produce a repeating seasonal cycle in the modeled phytoplankton community. We also looked at the trend in globally integrated annual average phytoplankton biomass, and found that it was in quasi-steady state (i.e. no identifiable trend) for the last few years of both simulations. The nutrient drifts that are linked to slow changes in the deep nutrient concentrations are small and significantly smaller than the seasonal and interannual variability.

We would also add that the model does not need to be in steady state for the resource competition theory framework to be useful. The model is never in exact steady state (we would argue that this is the case for all biogeochemical models). In this study we are examining the model results to ascertain whether or not the model solutions are behaving as we would expect them to under resource competition theory. Resource competition theory has previously been applied to a realization of this model (Dutkiewicz, 2009), and as in that study we demonstrate that the theory holds in the subtropics, where the most limiting nutrients are drawn down to the same concentration in both simulations, where the same modeled phytoplankton phenotype is also dominant. Dutkiewicz et al (2009) showed results from the 10[th] year of a (CR) simulation. In the work for that study the results from year 5, 10, 20 and 40 years were considered, and no discernable differences between years were found.

Given our previous studies, the stability of the ecosystem, and the lack of strong drift in nutrients, we do not feel we need further analysis on the issue of drift. However we appreciate the reviewers concern and will add a paragraph to the revised version of the manuscript to address these concerns: a more in depth detail of the model initial conditions (outlined above), a discussion of the stability of the ecosystem and mention of the tiny changes in nutrients due to drift.

Is annual averaging the best way to evaluate the similarities and differences that are seen in Figs. 1-5? As Figure 7 shows there are striking monthly differences at higher latitudes. Perhaps it would be more informative to compare and show the key physical,

ecological, biogeochemical properties in seasonal plots (e.g., winter, summer, fall, and spring)? Or maybe carefully selected Hövemoller type plots would be informative?

We do feel that the annual average plots are useful. However, we thank the reviewer for this useful suggestion. We plan to replace Figs. 6 and 7 in the revised manuscript with a Hovmoller diagram showing the seasonal evolution of the MLD and biogeochemical properties of the model (Fig. R1, below). This clearly shows differences in the seasonality of the two models.

I would be particularly interested in seeing if phytoplankton diversity and differences are more pronounced seasonally or during the progression of the spring bloom in the Atlantic.

This study does not address modeled phytoplankton diversity, so we do not see any need to add plots showing phytoplankton diversity in the annual mean or seasonally. However this would be an interesting topic for a future study.

In Figs. 1 and 2, it would be nice to see the CR results too.

As stated above in our response to Reviewer #2, we feel that adding plots of both sets of model output would add unnecessary extra figures. However, we would be happy to include the CR fields as supplementary material in the revised manuscript.

Fig. 3. I find that this lone figure made it difficult to really see the differences between the model configurations and found that I had to refer to Clayton et al., 2013 to really understand what was going on. This is somewhat frustrating, as some of this information seems to be necessary to understand the study. It would be really nice to have this all in one publication.

Fig. 3 shows the distribution of the dominant phytoplankton functional groups and phenotypes (which are analogous to species), so we are unsure why the reviewer felt the need to refer to the 2013 paper that was only concerned with phytoplankton diversity. The point of this figure is to show how strikingly similar the distributions of the dominant phytoplankton phenotypes and dominant functional groups are between the two models. The differences *should* be hard to see, as the model solutions are not all that different in terms of phytoplankton community structure (which is one of the main points of the paper).

We realize that for people not used to thinking about phytoplankton community ecology, section 3.2, which describes the phytoplankton community structure may be a little terse, so we will expand it in the revised manuscript. In order to understand the results showing the dominant phytoplankton phenotypes and functional groups, it is not necessary to refer back to Clayton et al (2013). We merely include the short paragraph (P5, L13-16) about the differences in biodiversity between the two models for completeness. The work in this manuscript does not address biodiversity patterns in the models, and it is not necessary to know about the patterns in biodiversity to understand any of the conclusions of this paper.

Fig. 4. What is the actual concentration? Is it realistic? I realize that the purpose of the study is not to figure out which is the 'best' simulation, but it would still be nice to see the absolute values.

We have included a figure (Fig. R2, below) in our response which shows the annual average surface nitrate and iron fields for the ECCO2 simulation as well as the

difference between the two models. The modeled concentrations are realistic. We would be happy to replace Fig. 4 with Fig. R2 in the revised manuscript.

Page 5 line 15 'tus' needs to be 'thus'

We will make the correction in the revised manuscript.

Section 4.2 (revised)

4.2. Subpolar regions.

[revised manuscript text omitted]

Figure R2. Top panel: Annual average surface concentrations of nitrate in mmol N m$^{-3}$ (left) and dissolved iron in mmol Fe m$^{-3}$ (right) in the HR model solution for model year 1999. Bottom panels: the difference in these properties between both models (HR - CR).

---

## Author Response (AR2)

**bg-2016-337**
**Biogeochemical versus ecological consequences of modeled ocean physics**
Sophie Clayton, Stephanie Dutkiewicz, Oliver Jahn, Christopher Hill, Patrick Heimbach, and Michael J. Follows

**Response to reviewers:**

We thank the two reviewers for their helpful comments on the revised manuscript. We have set out our responses to reviewer #3's specific comments below, in blue.

Reviewer # 3

Overall, I find the manuscript much improved from the last version. Some critical details, such as a better description of the model set-up, have been added and the explanations for the model differences are now much more concrete. However, I still think that the key messages of the paper could be improved. There is only a relatively brief discussion (2 short paragraphs) in the conclusion section on what the implications are and it would be nice if the authors spent more time on this. For example, what do their results mean for those developing the next generation of models? Where specifically should limited resources be directed? Can they recommend "optimal" model configurations for different types of investigations, e.g., biogeochemical vs. ecological focused studies, given that researches have limited computational resources and time? Surely this study has given the authors more insight into the implications than the rather generalized statements that they've written. If the key messages and implications can be articulated in a better manner, and some minor recommendations followed (see below), I would recommend publication.

We believe that it is beyond the scope of this paper to provide recommendations on "optimal" model configurations. Such recommendations would also need simulations with simple and more complex ecosystem components, and would indeed be driven largely by the questions addressed. Here rather we have identified the mechanisms which drive both the differences (in biogeochemistry) and the similarities (in ecosystem) between different physical setups. The utility of resource competition in understanding these differences and similarities is, we believe, a useful component for other users of biogeochemical/ecosystem models to understand the controls on their models. Our study has shown important implications of the control of physics in the results. We appreciate the reviewer pushing us to articulate these implications more completely and we do so with the following:

- The last few sentences of the abstract now read (P1, L13-17):
*"Although previous work has suggested that complex models may respond chaotically and unpredictably to differences in forcing and resolution, we find that our model responds in a predictable way to differences in ocean circulation, despite its complexity. The use of frameworks, such as resource competition theory, provide a tractable way to explore the differences and similarities that occur. As this model has many similarities to other widely used biogeochemical models that also resolve multiple phytoplankton phenotypes, this study provides important insights into how the results of running these models under different physical conditions might be more easily understood."*

- Last sentence of the Introduction (P3, L1-6):

*"We explore both the effect on the bulk biogeochemical properties and the community structure of the model solutions. The objective of this study is not to assess which model performs best with respect to reality or to suggest an optimal resolution. Rather, we aim to examine how changes in the resolution and parameterization of subgridscale processes of the model domain alter the emergent biogeochemical and ecological properties of this diverse ecosystem model, and specifically to identify tools to help understand these differences and similarities."*

- Additional sentences at end of the first paragraph of the Discussion (P7, L8-10):

*"Here we explore what drives these similarities and differences, frequently using simple theoretical constructs. The goal here is to provide a framework that other modellers can use to help understand some of the implications of the physical/biogeochemical and ecological consequences of different forcings and resolutions."*

- Additional sentence at the end of the first paragraph of the Conclusions (P11, L16-17):

*"We have used theoretical constructs to help us pull apart the reasons for these differences and similarities"*

- Additional paragraphs in the conclusions (P12, L6-20):

*"Our results show that the physics resolved by the models do matter, regardless of the scope of the question being addressed. The modeled MLD plays a central role in mediating bulk biogeochemical processes, specifically through vertical nutrient supply and the modulation of the light environment, which ultimately control the magnitude of bulk biological rate processes. Notably, although global values of primary production, and to a lesser extent phytoplankton biomass, are similar between models, the bulk biogeochemical properties of the model solutions differ regionally. This shows that whether one is interested in global or regional questions, model resolution is crucially important.*

*It may be tempting to conclude from the results presented here that, in fact, model resolution is less important when considering ecological problems. However, we would strongly caution against this, as although we have shown that the dominant constituents of the phytoplankton community remain largely unchanged regardless of model resolution, significant differences in phytoplankton productivity, diversity (Clayton et al., 2013) and overall community composition do result from differences in model resolution.*

*We have shown in this study, that even for complex ecological models, it is possible to explain differences in model solutions, and a similar approach could be taken to evaluate the effect of different model physics on different ecological model formulations."*

Specific comments:

Please include the CR results as supplementary material. It is essential to have the absolute values available somewhere for reference if only difference plots are in the main text.
We have a supplemental file that includes the following results from the CR simulation:
- Fig. S1 (a) annual average SST

- Fig. S1 (b) annual average MLD
- Fig. S2 (a) annual average phytoplankton biomass
- Fig. S2 (b) annual average primary production

We have also included the old Fig. 7 as Fig. S3 in the supplemental, as requested below.

I prefer Fig. R2 over Fig. 4. I would recommend that Fig. R2 be the one that appears in the published version.
We have replaced Fig. 4 with Fig. R2 in the revised manuscript.

In Section 3.1 (Pg. 5 line 17) can the authors include the globally integrated primary production rates for the models rather than just saying that they are "similar", i.e., make a quantitative comparison rather than a qualitative one.
We have added a more quantitative comparison of the globally integrated primary production rates for the models. The HR model has 0.6% higher annual primary production than the CR model, this has been added to P5 L16-17 in the revised manuscript.

Figure 6, the Hovmoller diagrams of the seasonal evolution of key model variables, is an improvement that makes it easier to understand section 4.2. However, I would recommend that the authors not throw out Fig. 7 from the original version, as it's nice to also see the comparison of monthly-integrated values. Perhaps this older figure can be kept as a supplemental figure?
We have included the old Fig. 7 in the supplemental, as stated above.